# A Novel Tiled Amplicon Sequencing Assay Targeting the Tomato Brown Rugose Fruit Virus (ToBRFV) Genome Reveals Widespread Distribution in Municipal Wastewater Treatment Systems in the Province of Ontario, Canada

**DOI:** 10.3390/v16030460

**Published:** 2024-03-17

**Authors:** Delaney Nash, Isaac Ellmen, Jennifer J. Knapp, Ria Menon, Alyssa K. Overton, Jiujun Cheng, Michael D. J. Lynch, Jozef I. Nissimov, Trevor C. Charles

**Affiliations:** 1Department of Biology, University of Waterloo, Waterloo, ON N2L 3G1, Canada; isaac.ellmen@stats.ox.ac.uk (I.E.); jenn.knapp@uwaterloo.ca (J.J.K.); r7menon@uwaterloo.ca (R.M.); akoverton@uwaterloo.ca (A.K.O.); jiujun.cheng@metagenom.com (J.C.); michael.lynch@metagenom.com (M.D.J.L.); jnissimov@uwaterloo.ca (J.I.N.); trevor.charles@uwaterloo.ca (T.C.C.); 2Metagenom Bio Life Science Inc., Waterloo, ON N2L 5V4, Canada

**Keywords:** Tomato Brown Rugose Fruit Virus (ToBRFV), ARTIC amplicon sequencing, lineage abundance estimation, wastewater

## Abstract

Tomato Brown Rugose Fruit Virus (ToBRFV) is a plant pathogen that infects important *Solanaceae* crop species and can dramatically reduce tomato crop yields. The ToBRFV has rapidly spread around the globe due to its ability to escape detection by antiviral host genes which confer resistance to other tobamoviruses in tomato plants. The development of robust and reproducible methods for detecting viruses in the environment aids in the tracking and reduction of pathogen transmission. We detected ToBRFV in municipal wastewater influent (WWI) samples, likely due to its presence in human waste, demonstrating a widespread distribution of ToBRFV in WWI throughout Ontario, Canada. To aid in global ToBRFV surveillance efforts, we developed a tiled amplicon approach to sequence and track the evolution of ToBRFV genomes in municipal WWI. Our assay recovers 95.7% of the 6393 bp ToBRFV RefSeq genome, omitting the terminal 5′ and 3′ ends. We demonstrate that our sequencing assay is a robust, sensitive, and highly specific method for recovering ToBRFV genomes. Our ToBRFV assay was developed using existing ARTIC Network resources, including primer design, sequencing library prep, and read analysis. Additionally, we adapted our lineage abundance estimation tool, Alcov, to estimate the abundance of ToBRFV clades in samples.

## 1. Introduction

### 1.1. Tomato Brown Rugose Fruit Virus Global Incidence, Impact, and Phylogeny

In Canada, fresh market tomatoes have been the most abundantly produced greenhouse vegetable since record keeping began in 1955. In 2022, >293 M kg of fruit with a farm gate value of >CAD 793 M were produced, with >213 M kg of fruit with a farm gate value of >CAD 540 M produced in Ontario [1]. The Tomato Brown Rugose Fruit Virus (ToBRFV) is a devastating plant pathogen that infects tomato and pepper plants and can reach nearly 100% disease incidence, causing massive crop losses and placing a high economic burden on farmers [2,3,4]. Thus, the control or elimination of this virus is a major obstacle in maintaining secure global tomato crop yields.

The first incidence of ToBRFV was reported by Salem and colleagues (2016) following a 2015 outbreak in Jordan. Shortly thereafter, Luria and colleagues (2017) identified ToBRFV as the causative agent of a 2014 tomato disease outbreak in Israel. It has spread to at least 35 countries since, including major tomato-producing countries such as China, Türkiye, and Mexico [4,5,6,7]. Global transmission and clade-specific mutations of ToBRFV strains are monitored through a curated NextStrain database updated by authorized curators with each updated version referred to as a build [8,9,10].

### 1.2. Taxonomic Classification, Distinguishing between Species, Lineages, and Strains

The ToBRFV is a member of the *Tobamovirus* genus, a group of highly virulent plant pathogens that includes 37 virus species with diverse and distinct host ranges [3,11,12]. The ToBRFV genome shares the highest sequence identity with Tobacco Mosaic Virus (TMV) isolate Ohio V at 82.4% shared identity, and fulfills species demarcation criteria [2,13]. All ToBRFV isolates available in NextStrain share greater than 90% sequence identity, meeting the demarcation criteria. The ToBRFV 2022 NextStrain build (version 3) contains 179 ToBRFV isolates, grouped into eight distinct clades which represent diverging lineages of ToBRFV strains [8,9].

### 1.3. ToBRFV Gene Content, Mode of Infection, and Host Immune-Escape

Tobamovirus virions are rod-shaped particles up to 300 nm long and 20 nm wide and are non-enveloped [11,14,15]. The genome structure and gene order are conserved amongst all tobamoviruses, consisting of a ~6.4 kbp monopartite single-stranded positive-sense RNA genome containing four open reading frames (ORFs). Replication proteins are encoded by ORF1 and ORF2, whereas virus structural proteins are encoded by ORF3 and ORF4 and expressed through subgenomic RNA [4]. ORF1 encodes a 126 kDa RNA-dependent RNA polymerase (RdRp), while ORF2 encodes a 183 kDa helicase protein through an amber UAG stop codon downstream of ORF1 [2]. ORF3 encodes a 30 kDa movement protein (MP) that facilitates cell-to-cell transmission through the plant plasmodesmata. The MP binds to multiple cellular components that transport the viral RNA through the plasmodesmata. Lastly, ORF4 encodes a coat protein that encapsulates the virus genome forming a thin rod-like structure [2,14].

Changes in the ToBRFV MP sequence break tobamovirus-resistance genes that have been introduced into most commercial crop species to provide protection against tobamoviruses [16]. ToBRFV MP amino acid mutations result in structural changes that prevent MP binding by the plant Tm-2^2^ receptor protein, thereby preventing activation of the hypersensitive response and leading to severe infection symptoms [16]. Recently, ToBRFV-resistant tomato seed lines have been commercially developed; however, only a limited number of varieties are currently available (https://www.vegetables.bayer.com/ca/en-ca/resources/news/bayer-launches-additional-commercial-varieties-with-intermediate.html, accessed on 16 March 2024; https://www.syngentavegetables.com/resources/ToBRFV, accessed on 7 June 2023).

### 1.4. The ToBRFV NextStrain Database and Phylogeny

The NextStrain database is a useful resource for scientists and policymakers to understand transmission routes and establish disease control measures [10]. For example, genomic sequence data from commercial seeds were analyzed using NextStrain phylogenies which alerted Dutch authorities to the illegal use of a ToBRFV cross-protective species [8]. ToBRFV genome sequences can be submitted and then incorporated into updated builds by authorized curators. This community-based approach has increased the number of genomes available in each version [8,9]. The ToBRFV 2019, 2020, and 2022 NextStrain builds contain 63, 118, and 179 ToBRFV strains, respectively, which can be phylogenetically grouped into three, six, and eight clades, respectively, representing diverging ToBRFV clades [8,9,10]. However, when Abrahamian and colleagues (2022) analyzed genome sequences of 123 ToBRFV strains retrieved from GenBank as well as another 22 isolated from plant tissues, they identified three distinct clades. Zhang and colleagues (2022) also phylogenetically analyzed 78 ToBRFV genomes and identified three distinct clades. These differences between identified clades are likely a result of the differing phylogenetic analysis tools and genome sequences used by each study. Additionally, the majority of isolated strains originate from the Netherlands and this sampling bias has likely skewed ToBRFV phylogenetic analyses [8,13].

To fully understand ToBRFV phylogeny, an increased number of genome sequences from various regions must be obtained, which can be aided by improved sequencing methods [4,13]. To aid in ToBRFV identification and surveillance efforts, we developed a robust and specific ToBRFV genome tiled amplicon sequencing assay utilizing short-read Illumina sequencing technology.

### 1.5. Use of PCR-Enrichment Sequencing Assays for Viral Detection

High-throughput shotgun sequencing can produce millions of reads from a single sample [17,18,19]. However, the low abundance of viral reads and the high abundance of host and background reads in environmental samples can make it challenging to sequence viral genomes without utilizing virus-specific enrichment methods such as PCR [18,20]. PCR enrichment assays are often used to amplify specific DNA sequences; however, they require the development of target-specific primers [21,22,23].

Recently, an open-source collaboration called the ARTIC Network has created adaptable user-friendly tools to develop virus-specific sequencing assays utilizing PCR enrichment (https://artic.network/; accessed on 20 February 2024). The ARTIC Network was originally developed for monitoring the transmission and evolution of viruses such as MERS-CoV and Ebola but has since been adapted to streamline the design of species-specific tiled amplicon sequencing assays [22,24]. ARTIC primer schemes are designed to target short overlapping fragments, referred to as amplicons, that span a specific genome of interest. PCR-enriched amplicons can be sequenced using either Illumina or Nanopore platforms to obtain complete genome sequences. ARTIC primer schemes have been used to target many viral genomes, including but not limited to Zika, Ebola, Dengue, and SARS-CoV-2 [22,24,25,26,27].

We utilized the ARTIC infrastructure to develop a nearly whole genome tiled amplicon sequencing assay that enriches ToBRFV genomes for Illumina sequencing. During our ongoing exploration of SARS-CoV-2 content in wastewater influent (WWI), we observed widespread occurrence of the ToBRFV sequences in samples from multiple treatment plants in Ontario, Canada (data available upon request). Indeed, several recent studies have observed ToBRFV sequences in WWI, indicating this virus is likely widespread and prevalent in wastewater systems in many countries [28,29,30].

### 1.6. ToBRFV Occurrence and Transmission in Wastewater

ToBRFV sequences have been observed in wastewater samples from Slovenia and California [28,29,30]. In 2020, Bačnik and colleagues demonstrated that wastewater influent contains active ToBRFV virions which, when concentrated, can cause asymptomatic infections in tomato plants.

Reclaimed wastewater is commonly used to irrigate crops, however several species in the *Tobamovirus* genus, including Pepper Mild Mottle Virus (PMMoV), remain transmissible in treated wastewater effluent [28]. Thus, ToBRFV could potentially be transmitted to crops via reclaimed wastewater during irrigation. Moreover, in 2023, Mehle and colleagues demonstrated that ToBRFV-infected plants shed active virions through their roots which can then spread to other plants through shared hydroponic systems that distribute nutrient solution. This suggests that plants exposed to ToBRFV-contaminated wastewater could potentially become infected and cause widespread outbreaks. This could be a challenge for growers who utilize reclaimed wastewater to irrigate crops [31,32]. However, it has not yet been shown that ToBRFV virions remain infective after conventional wastewater treatment methods. Thus, the possibility remains that treated wastewater effluent is suitable for irrigation and further work must be performed to determine ToBRFV transmissibility [28,29].

The entry mode of ToBRFV virions into wastewater has not yet been established. One possible explanation for ToBRFV transmission is through environmental sources such as waste from regional greenhouses and fields with ongoing ToBRFV infections [30]. Another possible mode of transmission could be through a human dietary route [30]. If the virus is present in produce and tomato- or pepper-based products, then the excretion of these products by humans could transmit ToBRFV virions into WWI [33]. To test these hypotheses, WWI could be obtained and tested for ToBRFV from regions without ToBRFV infections or where there is no tomato or pepper cultivation to eliminate the possibility of transmission through agricultural waste. Additionally, tomato and pepper products from grocery stores, as well as human feces, could be tested for the ToBRFV to establish a dietary route of transmission [33]. Similarly, a dietary mode of PMMoV transmission to human feces has been established, thus, the human dietary mode of ToBRFV transmission into WWI is the more probable scenario [33].

### 1.7. ToBRFV Detection and Sequencing

Current ToBRFV screening procedures, such as RT-qPCR and loop-isothermal amplification assays, can be used to determine the presence or absence of ToBRFV sequences, quantify the viral load, and provide more rapid results than sequencing assays. However, many of these assays cannot detect strain-specific mutations or they require the development of specialized strain-specific primers/probes to track transmission routes and determine sources of infection [4,29,34,35]. Moreover, increased evolutionary pressure from disease eradication efforts could result in the emergence of novel mutations that evade or reduce the effectiveness of current rapid presence/absence detection assays and eradication measures [13]. Thus, sequencing of complete ToBRFV genomes is essential to monitor virus transmission and evolution, predict emerging threats, and fully understand ToBRFV phylogeny [8]. However, high-throughput shotgun sequencing analysis is not ideal for the recovery of ToBRFV genomes due to the low recovery of viral sequences, high costs, and length of sample processing time [4]. To circumvent this problem, we designed a ToBRFV-specific tiled amplicon assay. Results show our tiled-amplicon assay provides a reliable cost-effective method to sequence ToBRFV genomes as well as aid in global surveillance efforts and uncover novel phylotypes of clades.

To aid in ToBRFV surveillance efforts, we utilized wastewater RNA extracts from five Ontario WWI collection sites to develop and validate a novel ToBRFV tiled amplicon sequencing (ToBRFV-Seq) assay. We prepared RNA sequencing libraries using our ARTIC-style ToBRFV-Seq assay and an RNA shotgun sequencing assay using the NebNext Ultra II RNA Prep kit. By evaluating the recovery of ToBRFV reads by each method, we demonstrate that our assay is a highly specific and efficient method for ToBRFV genome sequencing. Moreover, we adapted our in-house SARS-CoV-2 lineage abundance estimation tool, Alcov [36], to identify and estimate the relative abundance of ToBRFV clades. Our adapted tool, named Altob (https://github.com/Ellmen/altob; accessed on 20 February 2024), was able to analyze clade-specific mutations and estimate the abundance of ToBRFV NextStrain clades 1–4 and 6–8 in samples.

## 2. Materials and Methods

### 2.1. Viral RNA Extraction from Wastewater Influent

Wastewater influent (WWI) samples were collected in plastic bottles and shipped in coolers with ice packs from five wastewater treatment plants in Ontario, Canada (Appendix A). Samples from five sites (designated A–E) were processed using virion capture Nanotrap Microbiome A particles (Ceres Nanosciences, Manassas, VA, USA, #44202), then RNA was extracted using the Qiagen RNeasy Mini kit (Hilden, Germany #74104).

Briefly, bottles of WWI were inverted to mix and allowed to settle for 1 min at room temperature, then 10 mL of wastewater supernatant containing suspended biosolids was moved to a fresh tube. A negative control sample was prepared alongside extracts using 10 mL of nuclease-free DI water instead of WWI. The supernatant was mixed with 100 µL of ER2 solution by briefly vortexing. Then, 150 µL of Microbiome A particles were added and the samples were inverted to mix. The samples were then incubated at room temperature for 10 min, with three inversions after 5 min of incubation. Samples were placed on a 15 mL tube magnetic stand; the cleared supernatant was discarded and magnetic beads were retained.

Beads were gently resuspended in 1 mL of nuclease-free water and transferred to a sterile 1.5 mL microfuge tube. Beads were placed on a magnetic stand and incubated for 1 min and the supernatant was discarded without disturbing the bead pellet. Beads were resuspended 700 µL of RLT lysis buffer containing 1% 2-ME (BioShop, Burlington, Canada, MER002.500) by pipetting and then vortexing. Samples were incubated at room temperature for 10 min, then 600 µL of the supernatant from each sample was manually extracted using the Qiagen RNeasy Mini kit (Hilden, Germany, #74104). RNA was eluted in 80 µL of RNase-free water and the concentration of DNA and RNA was analyzed on a Qubit 4.0 fluorometer (Thermofisher Scientific, Waltham, USA, #Q33238). DNA was degraded using the Ambion DNase I kit (Invitrogen, Vilnius, Lithuania, #AM2222), samples were incubated with 4 units of DNase I at 37 °C for 45 min. RNA was recovered using the RNeasy Mini Kit RNA (Hilden, Germany #74104) clean-up protocol, DNA and RNA concentration were then determined by Qubit 4.0 analysis to ensure RNA integrity and complete removal of DNA.

### 2.2. Primer Design and Pooling

Primers were designed by first generating a multiple sequence alignment (MSA) from the genomes of 118 ToBRFV strains listed on the 2020 NextStrain Build (version 2), and sequences were obtained from NCBI (Appendix A). A consensus sequence was generated from the MSA and uploaded to the web-based tool PrimalScheme to generate primer sequences targeting overlapping amplicons of approximately 400 base pairs (bp) in length. PrimalScheme generated 20 amplicons for a total of 40 primers that cover 95.7% of the ToBRFV genome NC_028478.1, which was used for downstream read mapping analysis. Primers were synthesized by Integrated DNA Technologies. Primers amplifying odd and even numbered amplicons were diluted and mixed in two separate aliquots (pools 1 and 2, respectively) to avoid mispriming between overlapping amplicons (Figure 1). Each primer was pooled at a concentration of 0.5 µM in respective pools (Appendix A).

### 2.3. ToBRFV-Targeted Tiled Amplicon Sequencing Library Preparation

RNA extracts from WWI samples were reverse transcribed to synthesize cDNA using the LunaScript SuperMix RT kit (New England Biolabs, Ipswitch, MA, USA, E3010L); 10 µL of each RNA extract was used in a total reaction volume of 20 µL. First-strand cDNA synthesis products were PCR-amplified in two reaction mixtures containing primer pools 1 or 2 (Figure 1). The PCR was performed using Q5 2× Master Mix (New England Biolabs, Ipswitch, USA, M0492S), 6 µL of cDNA synthesis reaction, and 2.5 µL of one primer pool, adding ddH_2_O to a total volume of 25 µL. Samples were initially denatured at 98 °C for 30 s, followed by 35 cycles of 98 °C for 15 s, 63 °C for 30 s, and 72 °C for 30 s, ending with a final extension of 5 min at 72 °C.

PCR products were prepared for sequencing using the Illumina DNA Prep kit (San Diego, CA, USA, #20060060) using half of the manufacturer’s recommended volumes. Briefly, PCR products were purified using AMPure XP beads (Beckman Coulter, Brea, CA, USA, A63881), then assessed by Qubit 4.0 fluorometer and agarose gel electrophoresis. Purified PCR products were tagmented using on-bead tagmentation and barcoded with custom unique dual indexes using six PCR cycles. Indexed PCR products of ~400 bp were isolated using double-sided magnetic bead purification. The purified fragments were assessed using a Qubit 4.0 fluorometer and agarose gel electrophoresis.

### 2.4. RNA Shotgun Sequencing Library Preparation

RNA extracts from WWI were prepared for RNA sequencing using the NEB Ultra II RNA Sequencing kit (New England Biolabs, Ipswitch, USA, E7770S), following the manufacturer’s recommended protocol, Section 5. RNA extracts were quantified using a Qubit 4.0 fluorometer, then reversed-transcribed using first- and second-strand synthesis enzymes. AMPure XP beads were used to purify double-stranded cDNA fragments. Fragments were end-prepped by dA-tailing enzymes, then adaptor-ligated. Ligated fragments were purified using AMPure XP beads, PCR-enriched and barcoded with unique dual indexes (Illumina, San Diego, CA, USA, 20027213) using six PCR cycles. PCR products were purified using AMPure XP beads and then assessed using a Qubit 4.0 fluorometer and agarose gel electrophoresis.

### 2.5. Library Pooling and Illumina NextSeq Sequencing

ToBRFV-Seq and RNA shotgun-prepared libraries were pooled. Samples prepared using the same method, either RNA shotgun or ToBRFV-Seq, were pooled at equal concentrations and shotgun libraries were pooled at 15× the concentration of ToBRFV-Seq libraries. The pooled libraries were sequenced on an Illumina NextSeq2000 using 2 × 300 sequencing and P1 chemistry. Approximately 65% of the flowcell sequencing capacity was dedicated to these samples, for a total expected yield of 52 GB.

### 2.6. Read Processing, Taxonomic Classification, and ToBRFV Genome Alignment

Sequencing read quality and adaptor content were evaluated using FastQC v0.12.1 [37]. Low-quality reads and adaptor sequences were removed and reads were paired using Trimmomatic v0.39 using flags ILLUMINACLIP:<a>:2:30:7:2:TRUE, LEADING:3, TRAILING:3, SLIDINGWINDOW:5:15, MINLEN:125 [38]. RNA shotgun reads were trimmed using NebNext Illumina adaptor read sequence 1 AGATCGGAAGAGCACACGTCTGAACTCCAGTCA and read sequence 2 AGATCGGAAGAGCGTCGTGTAGGGAAAGAGTGT while ToBRFV-Seq reads were trimmed using the Illumina DNA Prep adaptor sequence CTGTCTCTTATACACATCT. The overall quality of read trimming was evaluated using FastQC v0.12.1 [37]. Read pairs passing quality control filters were taxonomically classified with Kraken2 v2.0.7-beta, using the flag-paired and the standard database, and reports of read taxonomic rankings were generated for each sample [39]. Quality-controlled read pairs were aligned to the ToBRFV genome NC_028478.1 using bowtie2 v2.3.5 with the flag-local [40]. Alignments were converted into bam files with samtools v1.9 files and anvio-7.1, and then visualized using Tablet 1.21.02.08 [41,42,43].

### 2.7. Altob Implementation and Synthetic Read Simulation

Our in-house SARS-CoV-2 variant of concern lineage abundance estimation tool, Alcov, was adapted to identify and estimate ToBRFV clades in Illumina paired reads [36]. We used Seaview to compute a multiple sequence alignment (MSA) with the MUSCLE algorithm using NC_028478.1 as the root genome and 124 unique genomes listed on NextStrain (Appendix A) [8,44,45]. Then, a phylogenetic Maximum Likelihood tree was generated using a GTR model and 100 bootstrap replicates and compared to the 2022 NextStrain ToBRFV phylogenetic tree (Appendix A). Phylogenetically grouped isolates were used to generate MSAs for each clade using NC_028478.1 as the root and the MUSCLE algorithm. MSAs were used to generate a list of mutations that appear in at least 50% of clade-specific isolates, which were then used to construct constellations of clade-specific mutations to estimate ToBRFV clade abundances using Altob. Positions in our mutation list are relative to position one of the reference genomes, whereas the mutations reported in the NextStrain phylogeny are relative to position one of their ToBRFV MSAs which is either 106 or 107 bp upstream of our start site.

Altob was adapted from our in-house tool Alcov by modifying the original mutation constellations and reference genome; all other functions and parameters were maintained. A detailed description of Alcov and its functions can be found in our preprint [36]. Briefly, Altob scans sites with a minimum coverage of 40 reads for the provided mutations and calculates frequencies as the number of mutations over the total reads at that site [36]. Altob computes the clade abundance which best explains the observed pattern of mutations utilizing linear programming, a well-studied optimization technique [36,46].

Synthetic read data used to benchmark Altob were generated using DGWSIM v0.1.13 and genomes of eight randomly selected ToBRFV strains, each belonging to a different NextStrain clade (Appendix A) [8,47]. Three datasets containing either 1000, 10,000, or 100,000 paired 2 × 250 Illumina reads were generated for each of the eight genomes without introducing random mutations or random DNA reads. Files containing 100,000 paired reads were combined to generate datasets with a mixture of clade-specific mutations. An equal number of reads from eight, four, and two genomes were combined, thus, each clade is represented by 12.5%, 25%, or 50% of reads, respectively. Altob is publicly available on GitHub (https://github.com/Ellmen/altob; accessed on 20 February 2024).

**Figure 1 viruses-16-00460-f001:**
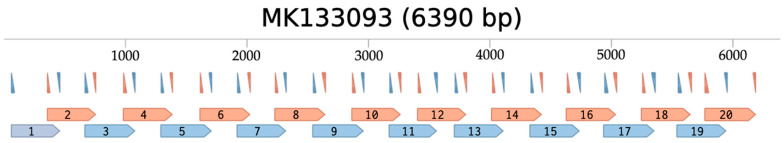
ToBRFV-specific primer binding sites and resulting amplicons mapped to the ToBRFV MK133093 genome sequence. In silico visualization of primer binding sites; forward and reverse primers are denoted by the upward and downward pointing triangles, respectively, and each resulting amplicon is numbered and denoted by blue and orange flags [48]. Pool 1 primers and amplicons are blue, pool 2 primers and amplicons are orange. Nucleotide positions are labeled at 1000 bp intervals.

## 3. Results

### 3.1. In Silico Primer Binding Analysis of ToBRFV Genome Sequences of 125 Strains

To ensure our 40 ToBRFV-specific primers would inclusively target a wide variety of ToBRFV strains without compromising species-specific stringency, we performed an in silico primer binding analysis using the sequence analysis software Benchling (2023) and 137 tobamovirus genomes [48]. A list of 179 ToBRFV strain accession numbers from the NextStrain ToBRFV build (2022, version 3) was obtained, representing 125 unique sequences, and these genomes were downloaded from NCBI. In addition, 12 tobamovirus RefSeq genomes of species closely related to the ToBRFV were obtained from NCBI (accessed on 20 April 2023) and all 137 unique genome sequences were imported into Benchling [12,15]. Benchling was used to probe all 137 genome sequences for potential primer binding sites using the program’s Find Primer Binding Sites tool. A successful match required at least 18 matching bases, a maximum of 3 mismatches, and a maximum of 1 consecutive nucleotide. The positions of matching primers were mapped to each genome and visualized using Benchling to ensure the overlapping amplicon primer scheme was maintained.

All 40 ToBRFV-specific primers had one matching binding site within each of the 125 ToBRFV-strain genomes (Table 1), with between zero and two mismatches observed between primers and ToBRFV-strain genomes. Several binding sites were identified in genomes of tobamovirus species closely related to the ToBRFV; 15, 14, 11, and 9 primers produced partial matches to tobacco mosaic virus (TMV), tomato mottle mosaic virus (ToMMV), rehmannia mosaic virus (ReMV), and tomato mosaic virus (ToMV) genomes, respectively (Table 1). Fewer matches, between zero and six, were identified for the other nine tobamovirus genomes which are more distantly related to the ToBRFV (Table 1) [2].

Our in silico primer binding analysis suggests all 40 primers could amplify the genomic sequences of all 179 ToBRFV strains listed on NextStrain while maintaining an overlapping binding scheme across each genome (Table 1, Figure 1). In silico results demonstrate that our primers are highly specific to ToBRFV genome sequences yet remain non-specific at the strain level. Moreover, limited binding to other tobamovirus genomes shows our primers are unlikely to amplify the complete genome of off-target species; however, partial genome segments could potentially be amplified.

### 3.2. Read Quality Control and Adaptor Trimming

Sequenced RNA shotgun libraries produced on average 4.64 Gbp and 15,627,313 reads per sample, whereas ToBRFV-Seq libraries produced on average 179,820 Mbp and 597,554 reads per sample, and in total 63.7 GB of data were produced (Table 2). Due to the low level of viral reads and high level of background eukaryotic and prokaryotic DNA found in environmental samples, shotgun-prepared libraries often require a high sequencing depth to obtain even a modest number of viral reads [18,49]. Thus, RNA shotgun samples were loaded at a 15× greater concentration than ToBRFV-Seq prepared samples. The majority of reads were high-quality, with RNA shotgun and ToBRFV-Seq prepared samples containing an average of 94.07% and 83.72% reads passing filters, respectively (Table 2).

### 3.3. Taxonomic Profile of WWI Samples Prepared by RNA Shotgun and ToBRFV-Seq Methods

Domain-level taxonomic classification of read pairs by Kraken2 against the standard database was computed to compare the recovery of viral reads obtained with each library prep method (Figure 2A,B) [39]. RNA shotgun reads were mostly bacterial, representing on average 89.40% of sample reads; 0.06% were viral, while the remaining 10.53% of reads were eukaryotic, archaeal, or other. Comparatively, on average, 97.38% of ToBRFV-Seq reads were classified as viral while the remaining 2.62% of reads were eukaryotic, viral, archaeal, or other (Figure 2C). The high recovery of bacterial reads in cDNA shotgun libraries is likely a result of residual bacterial genomic DNA, rRNA, and mRNA sequences in prepared samples and is a common limitation of viral metagenomic sequencing [50,51]. DNA contamination could be prevented by including thorough quality checks to confirm DNA has been completely removed from the sample prior to library preparation steps [51]. Notably, DNA contamination did not impact the recovery of viral reads using our ToBRFV-Seq, even though the same extracts were used for each library prep method. Moreover, we have found DNase treatment unnecessary to obtain a high number of specific reads with our ToBRFV-Seq assay (data available upon request).

Viral read pairs were further classified at the species level to evaluate the recovery of ToBRFV, other tobamovirus, and all other virus species by each library preparation method (Figure 2D,E). As expected, RNA shotgun sequencing yielded a mixture of tobamovirus reads with a high proportion specific to the ToBRFV, as well as other virus species (Figure 2D,F). On the other hand, our ToBRFV-Seq assay almost exclusively produced ToBRFV reads (Figure 2E,F). Although other tobamoviruses were observed in all WWI samples, they were not significantly enriched within ToBRFV-Seq samples (Figure 2G). This demonstrates that our primers do not amplify these closely related off-target sequences when present in low abundance and are therefore highly specific to ToBRFV genomes.

Of note, the ToBRFV-Seq blank sample, prepared using nuclease-free water, contained 100 ToBRFV read pairs (Figure 2G). This is likely due to sample contamination prior to or during library preparation and is a commonly encountered issue when preparing tiled-amplicon sequencing assays [52]. Thus, extra precautions should be taken to reduce sample contamination during RNA extraction and PCR setup [53].

### 3.4. Mapping Reads to a ToBRFV Reference Genome

The quality of ToBRFV genome recovery was assessed by mapping reads from samples to a ToBRFV genome using bowtie2, and assessing coverage, completeness, and depth of read mapping to the reference [40]. Our ToBRFV-specific primers target 95.7% of the ToBRFV genome sequence, forgoing 64 bp and 210 bp on the 5′ and 3′ genome ends, respectively. These regions are less than 400 bp and therefore too small to be included in our tiled amplicon primer scheme. Thus, the percentage of genome coverage was calculated for both the total and PCR-targeted genome regions.

Both library preparation methods were able to obtain nearly complete or complete coverage of the total and PCR-targeted genome; however, RNA shotgun samples contained far fewer reads and had a lower depth of coverage than ToBRFV-Seq samples (Table 3). Moreover, RNA shotgun samples on average contained 15,627,313 read pairs, yet only 0.02% of reads aligned to the ToBRFV reference genome (Table 2 and Table 3). Conversely, ToBRFV-Seq samples contained on average 597,554 read pairs with 99.86% of reads aligning to the reference genome (Table 2 and Table 3). RNA shotgun samples utilized a much greater sequencing capacity than our ToBRFV-Seq samples, yet still yielded fewer ToBRFV reads and lower depth of sequencing. Thus, our ToBRFV-Seq assay offers an alternative and more robust approach for sequencing ToBRFV genomes that is highly specific and requires minimal sequencing capacity. On average, ToBRFV-Seq samples covered 99.92% of the PCR-targeted genome sequence with 0–6 bp missing on alignment terminal ends.

### 3.5. Estimating the Relative Abundance of ToBRFV Clades in Wastewater Influent

We adapted our tool, Alcov, to estimate the abundance of ToBRFV clades in Illumina sequencing reads and named the adapted tool Altob. Clade-specific mutations were defined by constructing a Maximum Likelihood tree from unique ToBRFV isolates listed on the NextStrain 2022 build and clade-specific MSAs aligned to the reference genome NC_028478.1 were used to define mutations [8]. We were able to resolve seven of the eight clades identified on NextStrain; however, clade five isolates did not form a distinct clade and appeared in several divergent branches (Appendix A). Benchmarking analysis of Altob was performed using synthetic Illumina read datasets of 1000, 10,000, and 100,000 reads using genomes that represent each ToBRFV clade [8].

Altob correctly identified all clades and estimated between 94.2% and 98.7% relative abundance for the respective clades (Figure 3A), except for clade five. Clade five contains the wildtype ToBRFV used to root our tree and define mutations in all other clades; therefore, the clade five clade is not called by Altob [8]. Synthetic read datasets containing a mixture of clade-specific reads were also correctly identified using Altob (Figure 3B). We expected a 50% abundance call for each clade in files containing two genomes, a 25% call in files containing four genomes, and a 12.5% call in files containing all eight genomes. Slight differences in the estimated and expected read abundance for some clades were observed (Figure 3B), which can likely be attributed to the low frequency of mutations in some strains and therefore absence of mutations in some randomly generated reads. Thus, Altob is well suited to identify and distinguish between ToBRFV clades with the exception of clade 5; however, abundance scores should be treated as close estimates and not as exact measurements.

Finally, Altob was used to evaluate the clades present in our WWI samples (Figure 3C). In samples A–E, clades four and seven were identified in all RNA Shotgun and ToBRFV-Seq samples, demonstrating the wide-spread transmission of specific ToBRFV-clades in Ontario WWI. Clade four strains have been identified in North America; as such, the identification of clade four strains in Ontario wastewater influent is unsurprising [8]. Clade seven strains have been identified in the Netherlands and Belgium, suggesting the transmission of these strains to Ontario either through the agricultural industry and/or the import of tomato products [8,30,33]. In each sample, the predicted clade abundance of clade 7 was relatively similar regardless of library preparation method, with an average difference of ∓1.86% between RNA shotgun and ToBRFV-Seq estimates for each respective sample. However, clade 4 yielded different abundance calls when samples were prepared by each assay and had an average difference of ∓26.92%.

### 3.6. Assessment of Viral Shotgun Reads at a Species Level

To investigate the presence and abundance of viruses in WWI, viral reads from RNA shotgun samples were taxonomically classified at the species level and the abundance of species assigned read pairs was evaluated (Figure 4). In all five WWI samples, twenty-six viruses were consistently found which included plant-infecting viruses, human pathogens, enteric bacteriophages, insect viruses, and amoeba-infecting viruses (Appendix A).

The counts and percentage of paired reads assigned to the seven most prevalent virus species identified in each sample were compared to ascertain which species were consistently found in high abundance (Figure 4). In samples A–D, either ToBRFV or choristoneura fumiferana granulovirus (ChfuGV) read pairs were in the highest abundance while cucumber green mottle mosaic virus (CGMMV) and pepper mild mottle virus (PMMoV) were on average the third and fourth most prevalent virus species, respectively (Figure 4B). ToBRFV, CGMMV, and PMMoV are plant pathogens belonging to the Tobamovirus genus, while ChfuGV is an insect pathogen that infects budworms [3,54]. In sample E, however, Wuhan insect virus 23, is the most abundant single virus species detected. Preparation or sequencing biases may have resulted in the overestimation of this virus in samples. However, these samples were collected during summer months, therefore there may have been a high insect-to-human virus transmission experienced by individuals in the population serviced by site E. Replicative data at this site could be used to discern between these and other potential possibilities.

## 4. Discussion

ToBRFV-Seq is a whole genome sequencing enrichment assay that utilizes a tiled amplicon primer scheme for specific PCR amplification. We evaluated our ToBRFV-Seq assay specificity using RNA extracted from five wastewater influent samples. We assessed primer specificity by first performing in silico primer binding analysis to ToBRFV genomes and other closely related tobamovirus genomes (Table 1). In silico, all primers were capable of binding the genomes of all 125 ToBRFV strains that were analyzed while relatively few off-target binding sites to other tobamovirus genomes were identified. We then evaluated primer specificity by taxonomically identifying reads obtained from five WWI RNA extracts prepared with our ToBRFV-Seq assay. The vast majority were taxonomically identified as ToBRFV reads and very few reads, 0–0.0017%, were assigned to other tobamoviruses (Figure 1G). Furthermore, >99% of ToBRFV-Seq reads mapped to the reference genome NC_028478.1, covering >99% of the target genomic region and >95% of the total genome (Table 3). Thus, demonstrating our primers are highly specific to and efficient in enriching ToBRFV genomes. When evaluating our ToBRFV-Seq assay we also prepared each WWI RNA extract for RNA shotgun sequencing. ToBRFV was detected in all RNA shotgun samples; however, a far greater depth of sequencing was required to obtain a modest number of ToBRFV reads when compared to our ToBRFV-Seq assay (Table 3). Thus, our assay provides an effective sequence enrichment method that is highly specific to ToBRFV genomes.

The high relative abundance of ToBRFV-specific reads in RNA shotgun samples indicates that ToBRFV is widespread and highly prevalent in Ontario WWI (Figure 4). Moreover, comparing counts of ToBRFV-specific reads versus all other tobamovirus reads with a one-tailed t-test (*p* < 0.05, *n* = 5, df = 4, *t*-values, *t*-value = 3.569, critical *t*-value ±2.776) demonstrates a statistically significant difference in read counts, with ToBRFV reads outnumbering all other tobamovirus reads combined (Figure 2D). The higher abundance of ToBRFV reads than PMMoV and all other tobamoviruses is consistent with the findings of Rothman and Whiteson, 2022, who investigated WWI from Californian treatment plants for the presence of eight tobamoviruses using qPCR and Illumina sequencing. Of the eight tobamoviruses viruses, their analysis identified the ToBRFV as the most abundant in California WWI, even more prevalent than both PMMoV and CGMMV [30].

### 4.1. Possible Assay Applications to Determine ToBRFV Transmission and Prevalence

Wastewater influent has become an important tool in tracking the transmission of specific virus lineages [55,56]. Since the ToBRFV is prevalent in wastewater, sequencing of WWI could be used to track and identify currently circulating ToBRFV lineages. To this end, we adapted our SARS-CoV-2 lineage abundance estimation tool, Alcov, to estimate ToBRFV lineages from sequencing reads. Generally, Altob is a robust prediction tool capable of estimating ToBRFV clade abundances in samples with as few as 1000 ToBRFV reads (Figure 3). However, some differences in the estimated clades when using differing library preparation methods demonstrate that Altob-predicted clade abundances should be treated as general estimates. The identification of additional ToBRFV strain genome sequences, particularly in clades 6 and 8 which each contain only three known strains, would enable more clade-specific mutations to be identified and added to Altob constellation files, thus improving Altob’s ability to classify samples by ToBRFV clade. Together, our ToBRFV-Seq assay and Altob could be a valuable resource to identify clade-specific ToBRFV infections, explore potential transmission routes, and aid in combating transmission events.

PMMoV is a commonly used human fecal indicator due to its reliable presence in human feces and has become a popular RT-qPCR standard to normalize cycle threshold (Ct) values from SARS-CoV-2 assays [33,56,57,58]. Since the SARS-CoV-2 pandemic, a multitude of virus and chemical human fecal indicators, such as PMMoV, caffeine, and nicotine, have been recently evaluated [59,60,61]. However, a consensus on the best standard has not yet been reached and the reliability of PMMoV and other standards is quite variable between studies due to factors such as population size, seasonal changes, diet, sample processing, and detection methods [58,61,62,63,64,65]. Although PMMoV was found in all our RNA shotgun samples, it was consistently found in a lower abundance relative to ToBRFV, suggesting ToBRFV may be more readily detectable and thus be a more robust fecal indicator and RT-qPCR standard than PMMoV (Figure 4). Indeed, two studies examining human infant oropharyngeal and fecal viromes within the first year of life identified the ToBRFV by metagenomic sequencing in infant fecal [66] and infant oropharyngeal [67] viromes. Moreover, a recent comparative study of PMMoV and ToBRFV as fecal contamination indicators showed that ToBRFV is more prevalent in wastewater and matched the established microbial tracking marker cross-assembly phage (crAssphage) abundance [68]. This suggests that ToBRFV may be a suitable marker replacement for PMMoV in wastewater. ToBRFV has likely been overlooked as a fecal indicator because PMMoV was first introduced as a fecal indicator in 2006, long before the first recorded ToBRFV outbreak in 2014 [2,33,69].

### 4.2. Effectiveness of and Possible Improvements to Viral Capture and ToBRFV-Seq Procedures

Microbiome A particles, manufactured by Ceres Nanosciences™, are magnetic capture beads that offer a fast and easy solution to isolate numerous virion species from wastewater, including SARS-CoV-2, influenza, respiratory syncytial virus, PMMoV, norovirus, and others [55,70,71]. We utilized Microbiome A particles to recover ToBRFV sequences from all of our WWI samples (Figure 2) and to our knowledge, this is the first demonstration of ToBRFV capture and concentration using Microbiome A particles.

Virus sequence yields in samples concentrated with Microbiome A particles provide a rapid and user-friendly method to isolate ToBRFV and other viruses for targeted sequencing methods. However, Microbiome A beads may not be the best option for virus metagenomic RNA shotgun sequencing, due to the high contamination of bacterial DNA sequences. Thus, when performing viral metagenomic analysis, a combination of ultracentrifugation and filtration or other robust viral isolation methods should be utilized [72,73].

## 5. Conclusions

To aid in ToBRFV detection and ongoing surveillance efforts, we designed and developed a ToBRFV genome tiled amplicon sequencing assay that is robust, specific, and efficient. Moreover, we designed a computational tool, Altob, that estimates the relative abundance of ToBRFV clades in Illumina paired reads. Additionally, we demonstrate that Microbiome A particles are an effective method for isolating ToBRFV from WWI. Lastly, we demonstrate that ToBRFV is widespread and one of the most abundant virus species in Ontario, Canada wastewater influent. Due to its stability and abundance in WWI, the ToBRFV has the potential to be a useful indicator species in wastewater and/or used as an RT-qPCR standard to quantify viral content in wastewater and more broadly, used in other wastewater epidemiology efforts and human fecal studies; however, this would require further study [68]. Moreover, due to the transmissibility of several tobamovirus species in reclaimed wastewater, including PMMoV [28], the closely related ToBRFV may also be transmissible in reclaimed wastewater used in crop irrigation, which could facilitate transmission of active ToBRFV virions to crops causing viral outbreaks.

## Figures and Tables

**Figure 2 viruses-16-00460-f002:**
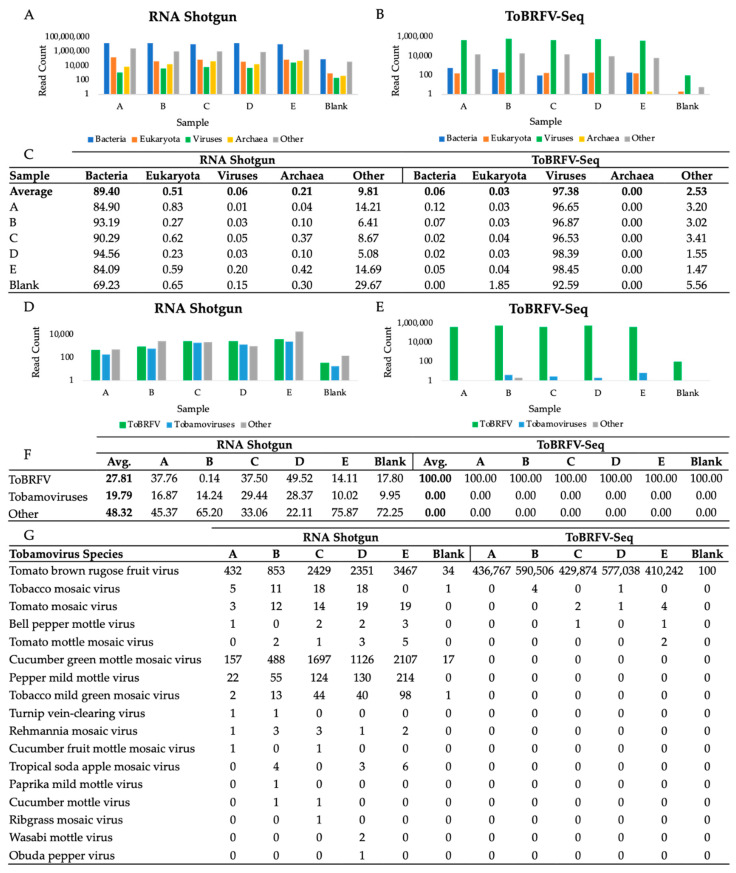
Taxonomic Classification of RNA Shotgun and ToBRFV-Seq Read Pairs at the Domain- and Species-level. Kraken2 classified read pair counts at the domain level for (**A**) RNA shotgun library prepared samples and (**B**) ToBRFV-Seq prepared samples [39]. At the domain level, categories included are viral, archaea, bacterial, eukaryotic, or other reads, where other includes unclassified reads, plasmids, adapters, linkers, and/or primers. (**C**) The percentage of reads represented by each domain for both prep methods. Samples A–E were used to calculate the average, the blank was excluded. (**D**,**E**) Percentage of total reads classified as ToBRFV (T), other tobamoviruses (OT), and the remainder of virus classified reads (RR) (**F**). Counts of tobamovirus classified read pairs at the species level (**G**). (**A**,**B**,**D**,**E**) Reads are represented on a log scale to magnify and visualize low abundance reads. Two-tailed tests were used to compare the mean read counts of RNA shotgun and ToBRFV-Seq classified reads, respectively. (**D**) RNA shotgun samples, *p* < 0.05, *n* = 5, df = 4, *t*-values, (T-OT) 3.569, (T-RR)-1.015, and (RR-OT)-1.181, critical *t*-value ± 2.776. (**E**) ToBRFV-Seq samples, *p* < 0.05, *n* = 5, df = 4, *t*-values, (T-OT) 12.523, (T-RR) 12.523, and (RR-OT) 2.449, critical *t*-value ± 2.776. Thus, there is a statistically significant difference between the number of ToBRFV and other tobamovirus reads produced by RNA shotgun sequencing, whereas there is a statistically significant difference between the number of ToBRFV reads produced compared to both other tobamovirus reads and the read remainder.

**Figure 3 viruses-16-00460-f003:**
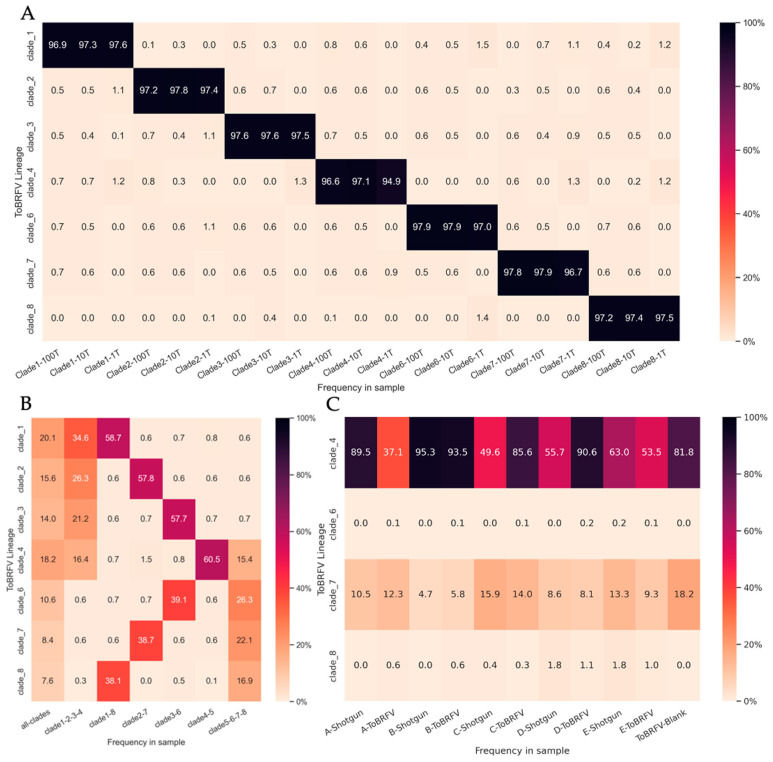
ToBRFV Clade-Abundance Estimates of Synthetic Read Datasets and WWI samples with Altob. Heatmaps display Altob clade abundance estimates of (**A**) datasets containing 1000, 10,000, or 100,000 synthetic reads from eight ToBRFV genomes representative of each clade. (**B**) Datasets combining 100,000 synthetic reads from either all eight genomes, four genomes, or two genomes. (**C**) Reads from WWI samples prepared with RNA shotgun (Shotgun) or our ToBRFV-Seq assay (ToBRFV) (A–E).

**Figure 4 viruses-16-00460-f004:**
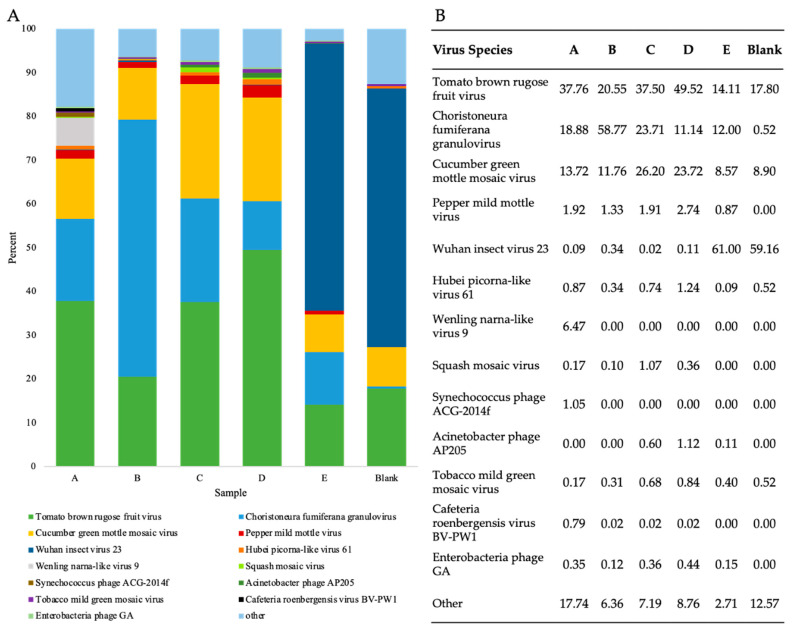
Evaluation of virus-species abundance in WWI samples using counts of viral reads taxonomically classified by Kraken2. (**A**) Percentage of virus paired read counts of the seven most abundant virus species found in each sample and count of all other species. (**B**) Percentage of virus species classified reads assigned to the most abundant species in each WWI sample prepared by RNA shotgun and a blank.

**Table 1 viruses-16-00460-t001:** In silico Analysis of ToBRFV-specific Primer Binding to 137 Tobamovirus Genomes. In silico primer binding analysis identified one potential binding site in 137 ToBRFV genomes for all 40 ToBRFV-specific primers [48]. A limited number of ToBRFV-specific primers are partially bound to other tobamovirus genomes. A greater number of primer binding sites was observed in genomes of species sharing more recent ancestry with the ToBRFV.

Tobamovirus Species	Accession	Primer Count
All 125 unique ToBRFV genomes	Appendix A	40
Tobacco mosaic virus (TMV)	NC_001367.1	15
Tomato mottle mosaic virus (ToMMV)	NC_022230.1	9
Tomato mosaic virus (ToMV)	NC_002692.1	14
Pepper mild mottle virus (PMMoV)	NC_003630.1	3
Bell pepper mottle tobamovirus (BPeMV)	NC_009642.1	6
Obuda pepper virus (ObPV)	NC_003852.1	1
Paprika mild mottle virus (PaMMV)	NC_004106.1	3
Tobacco middle green mosaic virus (TMGMV)	NC_001556.1	2
Cucumber green mottle mosaic virus (CGMMV)	NC_001801.1	0
Cucumber fruit mottle mosaic virus (CFMMV)	NC_002633.1	0
Brugmansia mild mottle virus (BrMMV)	NC_010944.1	6
Zucchini green mottle mosaic virus (ZGMMV)	NC_003878.1	0

**Table 2 viruses-16-00460-t002:** Total and Filtered Reads from RNA Shotgun and ToBRFV-Seq Prepared Sequencing Libraries. The total number of reads (Pairs), number of filtered paired reads (QF Pairs), and percent of quality filtered (%PF) reads obtained from WWI samples and a nuclease-free water blank prepared with either RNA shotgun or ToBRFV-Seq. Samples A–E were averaged, excluding the NTC (no-template control) blank.

	RNA Shotgun				ToBRFV-Seq			
Sample	Total Gb	Pairs	QF Pairs	% PF	Total Gb	Pairs	QF Pairs	% PF
Average	4.640000	15,627,313	14,631,264	94.07	0.1798200	597,554	502,007	83.72
A	5.5000000	18,528,523	17,138,798	92.50	0.1688000	560,908	451,928	80.57
B	4.9000000	16,434,198	15,693,587	95.49	0.2103000	698,848	609,572	87.23
C	3.6000000	12,035,815	11,788,803	97.95	0.1589000	527,913	445,326	84.36
D	5.4000000	18,266,390	16,248,968	88.96	0.2054000	682,673	586,488	85.91
E	3.8000000	12,871,637	12,286,164	95.45	0.1557000	517,430	416,720	80.54
Blank	0.0396000	131,614	123,528	93.86	0.0000571	190	108	56.84

**Table 3 viruses-16-00460-t003:** Alignment and Coverage of RNA Shotgun and ToBRFV-Seq Reads Mapped to a ToBRFV Genome. Number of ToBRFV reads and percentage of total reads that align to the ToBRFV genome NC_028478.1, % coverage of the total (% Total Cov) and PCR-targeted (% Target Cov) genomic region, the average read depth, and maximum read depth. Samples A–E were included in the average and the blank was excluded.

	RNA Shotgun						ToBRFV-Seq					
Sample	ToBRFV Reads	% Aligned	% Total Cov.	% Target Cov.	Avg. Depth	Max. Depth	ToBRFV Reads	% Aligned	% Total Cov.	% Target Cov.	Avg. Depth	Max. Depth
Average	3840	0.02	99.06	99.71	155.31	472	977,491	99.86	95.69	99.92	35,694.79	206,304
A	867	0.00	97.45	98.53	29.94	110	873,270	99.78	95.67	99.89	32,084.65	227,756
B	1732	0.01	99.34	100.00	67.63	184	1,180,695	99.85	95.70	99.85	44,893.64	312,224
C	4891	0.02	99.5	100.00	211.20	623	859,568	99.90	95.67	99.95	31,268.41	154,096
D	4742	0.02	99.44	100.00	177.90	556	1,153,735	99.90	95.74	99.95	40,645.85	227,468
E	6968	0.03	99.58	100.00	289.89	886	820,185	99.85	95.67	99.95	29,581.39	109,976
Blank	70	0.04	72.96	73.36	2.93	16	199	96.60	53.00	56.53	6.33	36

## Data Availability

NextStrain ToBRFV 2022 build access—https://nextstrain.nrcnvwa.nl/ToBRFV/20220412, accessed on 8 September 2023. Altob software access—https://github.com/Ellmen/altob, accessed on 20 February 2024. Raw read data with human reads removed available under BioProject PRJNA1030775.

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
