# Peer review of "A Novel Tiled Amplicon Sequencing Assay Targeting the Tomato Brown Rugose Fruit Virus (ToBRFV) Genome Reveals Widespread Distribution in Municipal Wastewater Treatment Systems in the Province of Ontario, Canada"

_viruses, 2024, doi:10.3390/v16030460_

Round 1

Reviewer 1 Report

Comments and Suggestions for Authors

The Tomato Brown Rugose Fruit Virus (ToBRFV) is a widespread plant pathogen that infects various Solanaceae crop species, leading to substantial losses in agricultural production. Consequently, controlling or eliminating this virus has become a primary concern.

Currently, screening procedures such as RT-qPCR, loop-isothermal amplification assays, and high-throughput shotgun sequencing are employed to monitor ToBRFV transmission and evolution, predict emerging threats, and comprehensively understand its phylogeny. However, these methods are constrained by the development of strain-specific primers/probes, low recovery of viral sequences, high costs, and extended sample processing times.

This study introduces a whole genome sequencing enrichment assay utilizing a tiled-amplicon primer scheme for specific PCR amplification, applied for detecting ToBRFV in wastewater RNA extracts from five Ontario WWI collection sites. Simultaneously, the authors compare their method with RNA shotgun sequencing to validate its reliability, specificity, efficiency, and cost-effectiveness in sequencing ToBRFV genomes. Additionally, the study proposes a computational tool, Altob, to estimate the relative abundance of ToBRFV-clades in Illumina paired reads. The research holds methodical significance by demonstrating the effective use of Microbiome A particles for isolating ToBRFV from WWI. While the work is interesting, well-designed, and well-written, some technical suggestions are provided for the authors to enhance the manuscript:

Remove the cited Fig. 1 in the introduction at lines 101 and 116.

Provide the full name of PMMoV at line 134.

Clarify that lines 175-185 are part of the results.

Correct "Fig. 1" to "Fig. 2" at line 406.

Consider adding parts C and F of Fig. 2 to supplementary data.

Label Fig. 4 with parts A and B.

Move lines 522-528 to the discussion section.

Correct the misspelling of "Figure" at line 540.

Correct the misspelling of "enriching" at line 543.

Addressing these points will strengthen the manuscript and provide clearer insights into the research findings.

Author Response

  • Figure 1 citation was removed from line 101 and 116.
  • full name of PMMoV was added to line 134
  • clarified that lines 175-185 are part of results
  • figure 1 was corrected to figure 2 at line 406
  • We considered moving figure 2 C and F to supplementary data. However, decided to keep these tables within our figure. Graphs A,B & D,E show the number of reads in each category, while C and F show the percentage of total reads in each category. Additionally, the read data in C and F are presented in graphs A,B, & D,E, respectively, and represented on a log scale, making it possible to visualize reads in low abundance categories.  This scaling however makes it more difficult for the reader to visually evaluate the percentage of reads belonging to each category. Thus, the tables C and F clarify the percentage of reads in each category. 
  • Labels A and B were added to figure 4
  • lines 520-531 were moved to the discussion section
  • misspelling of figure at line 540 was corrected
  • misspelling of enriching at line 543 was corrected

Reviewer 2 Report

Comments and Suggestions for Authors

The manuscript addresses an interesting and up-to-date scientific topic as the complete and deep sequencing of ToBRFV, which is recently spreading worldwide affecting tomato crops. Such sequencing data, together with a lineage abundance estimation tool also developed in this work, allow to identify and estimate the relative abundance of ToBRFV clades, tracking the evolution and points of new outbreaks from the prevalence in wastewater.

Although it seems quite a laborious and expensive assay for an epidemiology study in plant health, such strategy is undoubtedly sound and deserves to pave the way for other similar studies on other outbreaks, as it was already done for Sars-Cov2.

The manuscript describes the work with appropriate techniques, fitting figures and tables and is properly written. I suggest publication with two very small revisions:

-The name and acronym of the virus are not correctly written throughout the manuscript. I recommend to follow guidelines of ICTV for the taxonomy of viruses described at: https://ictv.global/faq/names 

Specifically, tomato brown rugose fruit virus (no capital letters and no italic) and ToBRFV (no italics).

- The captions of the first and second figure both report Figure 1, please correct.

Author Response

  • figure caption was corrected
  • ICTV nomenclature was reviewed and all instances of virus names and acronyms have been corrected. 

Reviewer 3 Report

Comments and Suggestions for Authors

The manuscript is well written and of scientific soundness. It provides a clear description of the methodology adopted, the obtained results and an appropriate discussion of these last, deserving of being published.

Authors are only invited to:

- review the orthography of the names of the viruses mentioned in the text and the related acronym according to ICTV rules;

- check the correct numbering of the figures (specifically, figure 1 is reffered to two figures whereas figure 2 is missing).

Author Response

  • ICTV taxonomy nomenclature was reviewed and updated throughout the manuscript
  • Figure number was corrected